# Hormone Receptor Expression in Meningiomas: A Systematic Review

**DOI:** 10.3390/cancers15030980

**Published:** 2023-02-03

**Authors:** Mikaël Agopiantz, Mélanie Carnot, Constance Denis, Elena Martin, Guillaume Gauchotte

**Affiliations:** 1Department of Assisted Reproductive Technologies, CHRU de Nancy, 54000 Nancy, France; 2Faculty of Medicine of Nancy, Université de Lorraine, INSERM UMRS 1256, Nutrition, Genetics and Environmental Risk Exposure (NGERE), 54000 Nancy, France; 3Department of Medical Gynecology, CHRU de Nancy, Université de Lorraine, 54000 Nancy, France; 4Department of Biopathology CHRU-ICL, CHRU de Nancy, 54500 Vandoeuvre-lès-Nancy, France

**Keywords:** meningioma, estrogen, progesterone, androgen, somatostatin, hormone receptors, recurrence

## Abstract

**Simple Summary:**

Meningiomas are, in most cases, low grade intracranial tumors. However, relapses are frequent. To date, only a few prognostic markers are described in the literature. Several studies have discussed the expression of progesterone, estrogen, androgen, and somatostatin receptors. The utility of analyzing these expressions for prognostic, theragnostic, and therapeutic purposes remains unclear. The aim of this study was to report the expression of these receptors, based on immunohistochemistry. Cochrane Collaboration guidelines and PRISMA statements were followed. We did an online search in PubMed using the MeSH database. References were selected if the investigations occurred from 1990 to 2022: 61 references were included. In this review, we describe the expression of these receptors in function of age, sex, hormonal context, localization, histological subtype, grade, and recurrence.

**Abstract:**

Meningiomas are, in most cases, low grade intracranial tumors. However, relapses are frequent. To date, only a few prognostic markers are described in the literature. Several studies have discussed the expression of progesterone, estrogen, androgen, and somatostatin receptors. The utility of analyzing these expressions for prognostic, theragnostic, and therapeutic purposes remains unclear. The aim of this study was to report the expression of these receptors, based on immunohistochemistry. Cochrane Collaboration guidelines and PRISMA statements were followed. We did an online search in PubMed using the MeSH database. References were selected if the investigations occurred from 1990 to 2022. 61 references were included (34 descriptive observational studies, 26 analytical observational studies, and one case report). In this review, we describe the expression of these receptors in function of age, sex, hormonal context, localization, histological subtype, grade, and recurrence.

## 1. Introduction

Meningiomas are the most common primary intracranial tumors, accounting for 36% of all intracranial tumors [1]. These tumors are well circumscribed and slow growing; approximately 75% are World Health Organization (WHO) grade 1. WHO histological grading is the most significant prognostic factor in meningiomas but, despite gross total resection, 7–25% of grade 1 meningiomas recur [2].

The hormone dependence of meningiomas has been demonstrated since the 1980s. Epidemiological data have been studied in several contexts. First of all, the meningioma is the only cerebral tumor whose predominance is female with a sex ratio estimated at 2:1 [3]. This ratio varies according to the period of genital activity of the woman: indeed, the incidence of meningiomas seems to increase after puberty, bringing the sex ratio to 3:1 in women of childbearing age. In addition, studies describe an increase in the size of known meningiomas at puberty, during the luteal phase of the menstrual cycle and during pregnancy [4,5,6]. In addition, an association between meningioma and breast cancer has been demonstrated. The risk of breast carcinoma after patients were diagnosed with meningioma was evaluated to 1.54 (95% confidence interval [95% CI], 0.77–2.75), and the risk of meningioma after patients were diagnosed with breast carcinoma was 1.40 (95% CI, 0.67–2.58) [7].

Epidemiological data were confirmed by histopathological data. Healthy meninges express progesterone receptors (PR) and between 33 and 89% of meningiomas express PR. There are three isoforms detected by Immunoblot: PR-A having a rather repressive effect in particular on the expression of PR-B, PR-B having a stronger affinity and a predominant activating action, and PR-78 [8]. They have heterogeneous expression, and PR-A is predominant in 66% of cases [9]. The expression of PR, which was the most studied, was previously reported to be inversely correlated with the tumor grade, as well as with the expression of the proliferation marker Ki-67 [10]. PR negativity is strongly associated with meningioma recurrence (*p* < 0.0001) [8], but there would be no correlation with recurrence-free survival in grade 1 meningiomas, R0 (total resection) [11].

The expression of estrogen receptors (ER) and androgen receptors (AR) has been less studied. ER expression is usually weak or even undetectable. A study has shown the mRNA expression of the two known isoforms (alpha and beta) of the receptor [12]. There is a positive correlation between the level of expression of ER and expression of Ki67 but with no clear correlation with grade [13,14]. ER expression was reported to be an independent factor of poor prognosis for progression-free survival and overall survival, particularly in grade 3 meningioma [15,16]. AR expression appears variable, most often found in more than 50% of meningiomas, and would be greater in women than men [17]. Androgen receptor binding activity was positively correlated with the progesterone receptor binding activity (RS = 0.38, *p* < 0.05) [18].

Furthermore, the risk of developing a meningioma is greatly increased with progestogen treatments. It was shown to be multiplied by seven after 6 months of treatment with cyproterone acetate and by 20 after 5 years of treatment, by 12 after 5 years of treatment with nomegestrol acetate, and by seven after 3.5 years of treatment with chlormadinone acetate [17]. The natural history of meningioma in this context shows a preferential localization at the anterior base of the skull, with more than 60% of multifocal forms. PR expression would be more intense compared to unexposed meningiomas, regardless of tumor location. In these cases, grade 1 was reported to be very prevalent (90%) with a cystic component in more than half of the cases under cyproterone acetate [19]. Finally, the mutational profile showed the lower presence of NF2 mutations and loss of function of chromosome 22 (7.5% versus 32%, *p* < 0.001), and more mutations of PIK3CA and TRAF7 [20].

It is on this basis that mifepristone, a powerful anti-progestogen, was used in a therapeutic trial and unfortunately showed no statistical difference between the arms in terms of progression-free nor overall survival between mifepristone and placebo in cases of unresectable meningioma in addition to radiotherapy [21]. Similarly, a clinical trial using a SERM found neither overall or progression-free survival (HR = 1.2, 95% CI [0.89–1.62], *p* = 0.239) to be significantly improved by the tamoxifen treatment [22]. A precise stratification of the patients based notably on the hormonal receptors’ expression profile could potentially improve the efficiency of anti-hormonal treatment for patients with inoperable relapsing meningiomas. However, the available data in the literature about hormonal receptors expression is heterogeneous and based on different analytic methods.

Additionally, among the other hormone receptors, somatostatin receptors (SR), and most notably SSTR2A, have been studied as markers and targets for the diagnosis and treatment of meningioma [23,24,25].

The objective of this study was to synthesize the epidemiological, clinical, and histopathological data of meningiomas of any setting according to the expression of the hormone receptors PR, ER, AR, and SR in a physiopathological and prognostic aim, using a systematic review. We will also discuss theragnostic aspects and the antihormonal therapeutic perspectives.

## 2. Materials and Methods

This systematic review was conducted according to the guidelines of the Cochrane Collaboration and the Preferred Reporting Items for Systematic Reviews and Meta-analyses (PRISMA) [26]. We did an online search in PubMed using the MeSH database. The MeSH terms “meningioma”, “progesterone receptor”, “estrogen receptor”, “androgen receptor”, and “somatostatin receptor” were used. Thus, our search was as follows: (meningioma[MeSH Terms]) AND ((progesterone receptor[MeSH Terms]) OR (meningioma[MeSH Terms]) AND (estrogen receptor[MeSH Terms]) OR (meningioma[MeSH Terms]) AND (androgen receptor[MeSH Terms]) OR (meningioma[MeSH Terms]) AND (somatostatin receptor[MeSH Terms]).

This systematic review has been registered in the PROSPERO database (ID 388909).

Publications from 1990 to March 2022 were studied. The last search on these databases was performed on March 1, 2022. Studies were included if they were investigating the clinical (recurrence, survival) and histopathological features (localization, histological grade) of the meningiomas with data on hormonal receptors immunohistochemical expression (PR or ER or AR or SR) in surgical specimens. Studies were excluded if full text was not available, if the publication language was not English or French, if they were not related to the topic, or if they were letters, comment, or narrative reviews. We excluded studies without a description of the population, as well as studies without strong data on hormonal receptors expression, studies with mRNA expression, no quantitative data on receptors expression, or without threshold of positivity. A PRISMA flowchart was made including reasons for exclusion of the literature (Figure A1). A total of 61 articles were included [8,10,11,12,13,15,16,17,23,24,27,28,29,30,31,32,33,34,35,36,37,38,39,40,41,42,43,44,45,46,47,48,49,50,51,52,53,54,55,56,57,58,59,60,61,62,63,64,65,66,67,68,69,70,71,72,73,74,75,76,77].

Data were extracted by a unique investigator in this sequence: reference, design of the study, patients, intervention, clinical and biological results, and adverse effects if reported.

Risk of bias was assessed for each study by different tool depending on the study design. The evaluation of the risk of bias of the descriptive and analytic observational cross-sectional studies were carried out using the modified Newcastle Ottawa Scale (NOS) Quality Assessment Scale. This scale was adapted for cross sectional studies [78] to provide quality assessment of cross sectional studies, which judges the relevance and the methodology of the review as well as the quality of the statistical analysis. Concerning the retrospective observational studies, we used the NOS for cohort studies.

Due to the heterogeneity of the included studies, meta-analysis could not be done. We summarized the effect estimated by combining *p*-values, since the type of data and statistical methods and tests varied. We presented the results with harvest plot with characteristics of the studies, represented by using different heights and shading. The confidence intervals in the results were measured by Wilson confidence interval.

## 3. Results

### 3.1. Risk of Bias

Most of the studies found were at medium (28 references) or high (34 articles) risk of total bias (Appendix A).

### 3.2. Hormone Receptor Expression in Meningiomas

72.2% of meningiomas expressed progesterone receptors (95% CI: 67.6–76.8). This value was calculated from the data of 43 articles (Figure 1).

Twenty-six articles allowed the study of the expression of estrogen receptors in meningiomas (Figure 2): these receptors were expressed in 11.3% of cases (95% CI: 5.9–16.7).

After analysis of data from seven related articles, androgen receptors (AR) expression was found in 45.5% of meningiomas (95% CI: 23–68). These results are presented in Figure 3.

### 3.3. Hormone Receptor Expression in Meningiomas According to Hormonal Status

#### 3.3.1. Gender and Hormonal Context

78.5% of meningiomas developed in female patients expressed progesterone receptors (95% CI: 73–84) versus 65.1% of meningiomas in males (95% CI: 57–73.2). These results were obtained after analysis of 32 and 25 articles, respectively. Concerning the hormonal context, 78.2% of meningiomas developed in a pre-menopausal context expressed progesterone receptors (95% CI: 61.9–94.5) vs. 68.4% of meningiomas developed in a post-menopausal context (95% CI: 62.6–74.2). These results were obtained after analysis of six and two articles, respectively.

After analysis of data from 20 and 16 articles, respectively, 14.7% of meningiomas developed in female patients expressed estrogen receptors (95% CI: 5.9–23.5) compared with 7.5% of meningiomas developed in males (95% CI: 0.9–14.1).

Finally, androgen receptors were expressed in 72.1% of meningiomas developed in women (95% CI: 48.8–95.4) and in 50.6% of meningiomas developed in men (95% CI: 37.5–63.7). Data were based on the analysis of four and three articles, respectively.

These results are presented in Figure 4.

#### 3.3.2. Age

The expression of progesterone receptors in meningiomas tends to decrease with age (Figure 5).

#### 3.3.3. Pregnancy

100% of meningiomas developed in pregnant or postpartum patients expressed progesterone receptors compared to 75.7% of women not meeting these criteria. These results were obtained after analysis of five and 27 articles, respectively (Figure 6).

Estrogen receptors were expressed in 20% of the cases of meningiomas developed in these first cases (95% CI: 0–44.9) against 14.9% in non-pregnant women (95% CI: 5.2–24.6). These results were obtained after analysis of five and 18 articles, respectively (Figure 6).

#### 3.3.4. Hormonal Treatment

89% of meningiomas developed under hormonal treatment expressed progesterone receptors (95% CI: 78.7–99.3) versus 71% of meningiomas developed without any hormonal treatment (95% CI: 66.3–75.7). These results were obtained by analyzing data from three and 41 articles, respectively (Figure 7).

Whatever the grade, meningiomas under hormonal treatment tended to express more progesterone receptors (75.4% versus 93.5% for grade 1 meningiomas and 57.2% versus 91% for grade 2 meningiomas). The results were obtained after analysis of 24 and two articles for grade 1 meningiomas and 15 and two articles for grade 2 meningiomas, respectively (Figure 8).

After analysis of the data from two and 24 articles, respectively, 6% of meningiomas developed under hormonal treatment expressed estrogen receptors (95% CI: 0.1–11.9) vs. 11% of meningiomas developed without any hormonal treatment (95% CI: 5.9–17.5). These results are presented in Figure 7.

### 3.4. Hormone Receptor Expression in Meningiomas According to Localization and Histological Features

#### 3.4.1. Localization

73% of intracranial meningiomas expressed progesterone receptors (95% CI: 67.9–78.1) compared with 69.9% of spinal meningiomas (95% CI: 45.3–94.5). Analysis of data from 36 and seven articles provided these results. Skull base meningiomas expressed progesterone receptors in 84.5% of cases versus 74.1% of non-skull base meningiomas (95% CI: 73.5–95.5 and 95% CI: 57.2–91). These results were obtained after analysis of data from 10 and eight articles (Figure 9).

Regarding estrogen receptors, 11.9% of intracranial meningiomas expressed the latter versus 15% of spinal ones (95% CI: 6.1–17.7 and 95% CI: 0–44.4). These data were obtained after analysis of 24 and two articles (Figure 9), respectively.

#### 3.4.2. Histological Subtype

81.1% of meningothelial meningiomas expressed progesterone receptors (95% CI: 75.5–86.7) compared to 75.8% of transitional meningiomas (95% CI: 68.5–83.1), 58.1% of fibroblastic meningiomas (95% CI: 45.8–70.4), 81.4% of psammomatous meningiomas (95% CI: 68–94.8), 87.5% of secretory meningiomas (95% CI: 63–100), 50.5% of atypical meningiomas (95% CI: 31.8–69.2), and 38.5% of anaplastic meningiomas (95% CI: 19.8–57.2). These results were obtained after analysis of data from 23, 22, 20, eight, eight, 15, and 18 articles, respectively. The latter are presented in Figure 10.

#### 3.4.3. Grade

76.8% of grade 1 meningiomas expressed progesterone receptors compared to 61.2% of grade 2 and 17.3% of grade 3 meningiomas (95% CI: 71.2–82.4, 95% CI: 48.2–74.2, 95% CI: 4.8–29.8). 26, 17, and 14 articles, respectively, dealt with this subject.

Estrogen receptors were found in 8.7%, 1.6% and 6.8% of grade 1, 2, and 3 meningiomas, respectively.

There was no difference in androgen receptor expression according to grade.

These results are presented in Figure 11.

#### 3.4.4. Recurrence

72.8% of primary meningiomas expressed progesterone receptors compared with 53.2% of recurrent meningiomas (95% CI: 64–81.6 and 95% CI: 33.6–72.8). These data were obtained after analysis of 11 and nine articles. Concerning estrogen receptors, these were found in 22.0% of primary meningiomas and 22.6% of recurrent meningiomas, after analysis of eight articles (95% CI: 11.2–32.8 and 95% CI: 0–46.4). These results are presented in Figure 12.

### 3.5. Focus on Somatostatin Receptor Expression in Meningiomas

The SSTR-2/SSTR-2a subtype is the most frequently expressed in meningiomas. On average, 87.6% of meningiomas expressed this latter (95% CI: 79.1; 96.1) versus 54% for the SSTR-1 subtype (95% CI: 0; 100), 57.1% for the SSTR-3 subtype (95% CI: 24.5; 89.7), 60.3% for the SSTR-4 subtype (95% CI: 26.8; 93.8), and 63.1% for the SSTR-5 subtype (95% CI: 12.2; 100). These results are presented in Figure 13 and were obtained after analysis of eight and three articles for the SSTR-2/SSTR-2a subtype and for the others, respectively.

Figure 14 summarizes the results of analyses comparing the expression of this SSTR-2/SSTR-2a subtype in meningiomas according to different parameters. 67.7% of meningiomas developed in female patients expressed somatostatin receptor SSTR-2/SSTR-2a subtype (95% CI: 59.7; 75.7) versus 77.4% of meningiomas in males (95% IC: 48.2; 100). These results were obtained after analysis of two articles.

74.4% of grade 1 meningiomas expressed somatostatin receptor SSTR-2/SSTR-2a subtype compared to 94.7% of grade 2 (95% CI: 47.9–100 and 95% CI: 84.2–100). Three articles dealt with this subject.

Concerning the histological subtypes, 68.2% of meningothelial meningiomas expressed somatostatin receptor SSTR-2/SSTR-2a subtype (95% CI: 43.8; 92.6) compared to 59.2% of transitional meningiomas (95% CI: 1.3; 100), 58.3% of fibroblastic meningiomas (95% CI: 50.1; 66.5), and 82.7% of atypical meningiomas (95% CI: 63.8; 100). These results were obtained after analysis of data from three articles.

## 4. Discussion

As expected, the vast majority of meningiomas express progesterone receptors in almost three quarters of cases. Less than half express androgen receptors and a very small part express estrogen receptors (11.3%). The proportion of cases which progesterone receptors are present is greater in some situations: 1/female gender; 2/the period of genital activity in women; 3/during pregnancy or postpartum; and 3/under hormonal treatment.

These results are less clear for estrogen, androgen, or somatostatin receptors. These situations correspond to a hormonal climate with significantly higher serum concentrations of estradiol and progesterone. These very interesting results should be linked to the various studies showing an increased risk of developing a meningioma under progestogen hormone treatment, particularly after exposure to cyproterone acetate. The latest data show an increased risk associated with the duration of treatment, with the cumulative dose, of meningiomas whose volume decreases after cessation of exposure to progestins in 79% of cases [79]. There are thus well-established data showing that these molecules capable of binding to progesterone receptors promote the development and progression of meningiomas.

Paradoxically, the presence of progesterone receptors is associated with significantly more favorable prognostic factors. Thus, grade 1 meningiomas express progesterone receptors in more than three quarters of cases, while grade 3 meningiomas express it in less than 20%. Similarly, recurrent meningiomas express significantly less progesterone receptors. These results are not clearly found for estrogen and androgen receptors. However, several studies in the literature show a negative correlation between survival and the presence of estrogen receptors. Moreover, no risk association was observed with drugs interacting with androgen levels such as GnRH agonists or anti-androgen hosts [80].

The vast majority of meningiomas are grade 1 benign tumors, but 15% are atypical and 2% are anaplastic according to the histological criteria of the WHO 2021. These high-grade meningiomas, like the recurrent nature of the pathology or the difficulties due to incomplete, repeated or deteriorating surgical resections, are all situations where the combination of surgery and radiotherapy leaves a therapeutic vacuum. No systemic molecule has currently proven itself in these situations [81]. This is how the use of targeted therapies, in particular antiangiogenic or inhibiting the mTOR pathway [82,83,84], have been proposed with mixed results.

Equally mixed results have been found with hormonal therapies [85]. A recent meta-analysis published in 2020 combining this study, included a randomized phase 3 trial showing that mifepristone did not show major efficacy [80]. It is very likely that antiprogestin agents can address a fraction of the population with an efficacy that will then be considered significant. The fact that progestins have been shown to be a factor in the progression of the disease capable of refluxing when they are stopped, while being associated with fewer recurrences and better survival, is quite in favor of this hypothesis.

Finally, although some controversies still remain and warrant further studies, accumulating evidence showed that SSTR2-related/targeted treatments (e.g., somatostatin analogs and SSTR2-targeted peptide receptor radionuclide therapy (PRRT)) are promising and safe therapeutic options for unresectable or refractory meningiomas [23,25,86,87].

## 5. Limitations of This Review

This systematic review had several limitations: problem of homogeneity of studies; survival data could not be analyzed; few studies focused on the androgen receptor; absence of consideration of hormonal treatments in the analysis of the population; lack of data about the precise tumor localization; and intra-tumoral heterogeneity with potential sampling bias. Moreover, the immunochemistry methods used to investigate the different receptors has changed over a 30-year period, and in particular the antibodies have improved. Therefore, for these reasons as well as to maintain a relative homogeneity, the oldest studies (before 1990) were excluded and we kept only the studies evaluating the expression of receptors by immunohistochemistry. However, a negative immunostaining does not necessary mean that the receptor is not present. Prospective studies are needed in order to confirm these results.

## 6. Conclusions

This systematic review has synthesized current knowledge based on the available data about the expression of receptors for the three main steroid hormones and somatostatin, basing on immunohistochemistry, which is a costless and easy-to-use method in daily practice. The data concerning the progesterone receptor are currently the strongest. Data on estrogen and androgen receptors are still patchy and require further study. The analysis of the expression of hormone receptors in meningiomas, therefore, appears to be necessary for future pre-therapeutic analysis in targeted hormonal therapy, for which a target reference population needs to be defined in large prospective clinical studies.

## Figures and Tables

**Figure 1 cancers-15-00980-f001:**
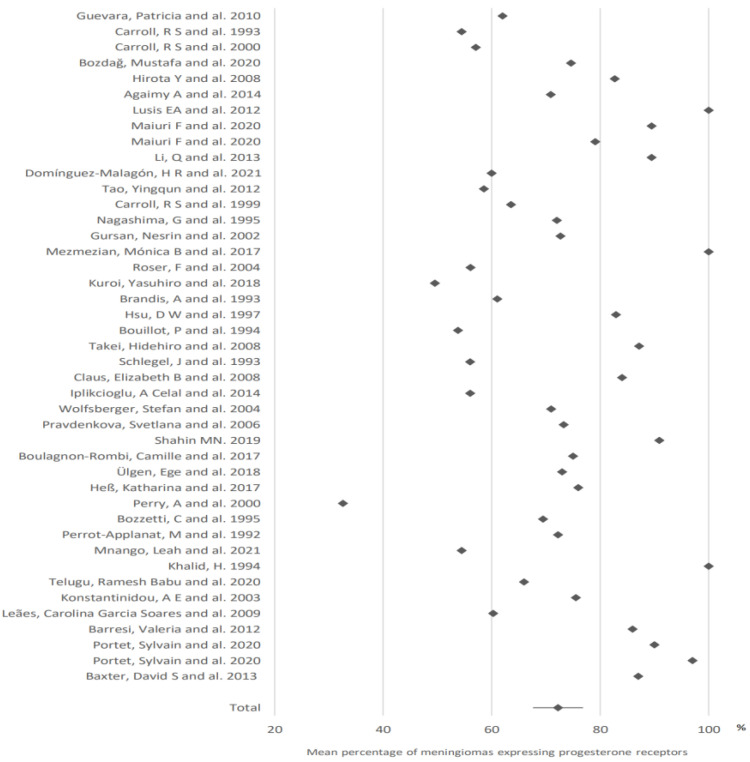
Expression of progesterone receptors in meningiomas. The results are expressed as a mean associated with their 95% confidence interval. Number of articles analyzed related to the topic: 44. Number of included cases (total for all studies; minimum; maximum in each study): 3918 (11; 588).

**Figure 2 cancers-15-00980-f002:**
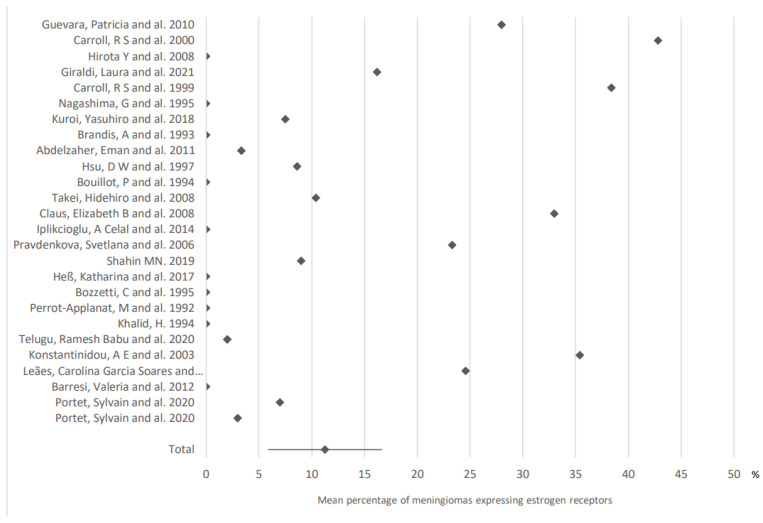
Expression of estrogen receptors in meningiomas. The results are expressed as a mean associated with their 95% confidence interval. Number of articles analyzed related to the topic: 26. Number of included cases (total for all studies; minimum; maximum in each study): 1528 (21; 161).

**Figure 3 cancers-15-00980-f003:**
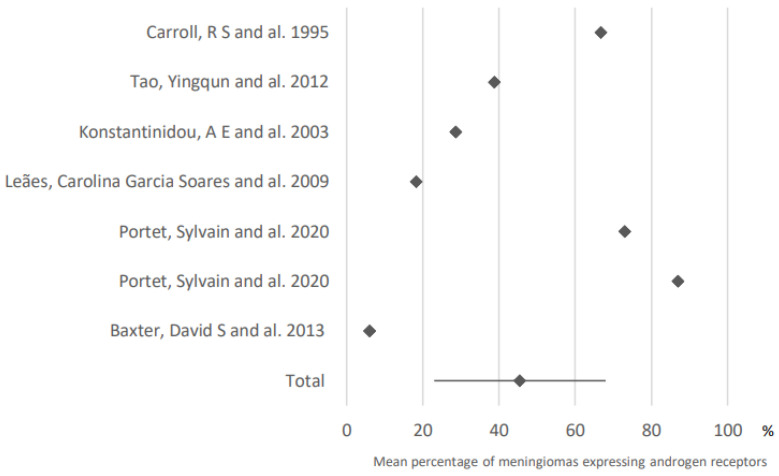
Expression of androgen receptors in meningiomas. The results are expressed as a mean associated with their 95% confidence interval. Number of articles analyzed related to the topic: 7. Number of included cases (total for all studies; minimum; maximum in each study): 613 (30; 175).

**Figure 4 cancers-15-00980-f004:**
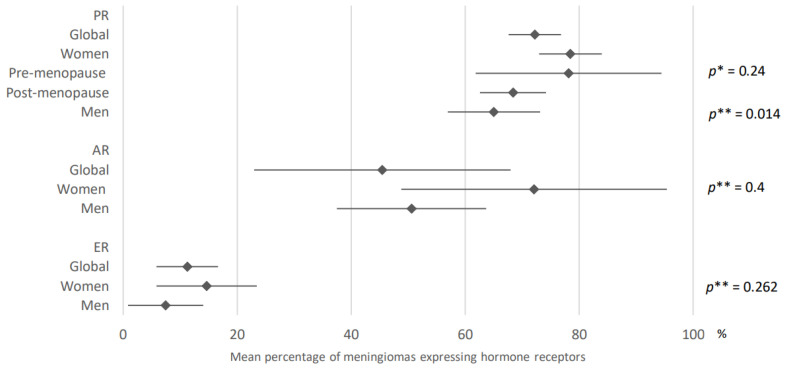
Comparison of hormone receptors expression in meningiomas according to gender. PR: progesterone receptors; AR: androgen receptors; ER: estrogen receptors. The results are expressed as a mean associated with their 95% confidence interval. *p**: pre-menopause versus post-menopause, *p***: men versus women. Number of articles analyzed related to the topic: Progesterone receptors: Women: 32; Pre-menopause: 6, Post-menopause: 2, Men: 25; Androgen receptors: Women: 4, Men: 3; Estrogen receptors: Women: 20, Men: 16; Number of included cases (total for all studies; minimum; maximum in each study): Progesterone receptors: Women, 1940 (8; 396); Pre-menopause, 160 (10; 68); Post-menopause, 33 (7; 26); Men, 747 (3; 192); Androgen receptors: Women, 438 (21; 357); Men, 105 (1; 86); Estrogen receptors: Women, 1036 (15; 357); Men, 341 (1; 86).

**Figure 5 cancers-15-00980-f005:**
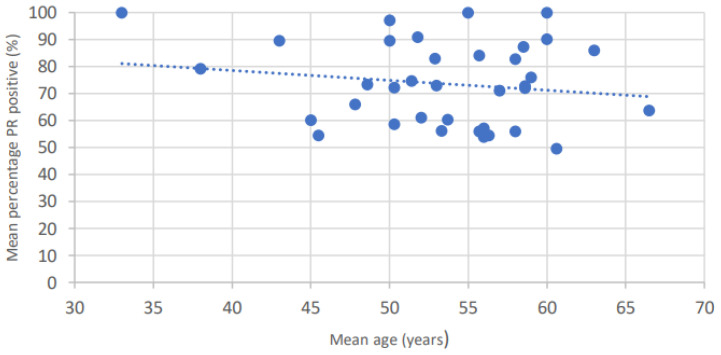
Age-dependent changes in progesterone receptors expression in meningiomas.

**Figure 6 cancers-15-00980-f006:**
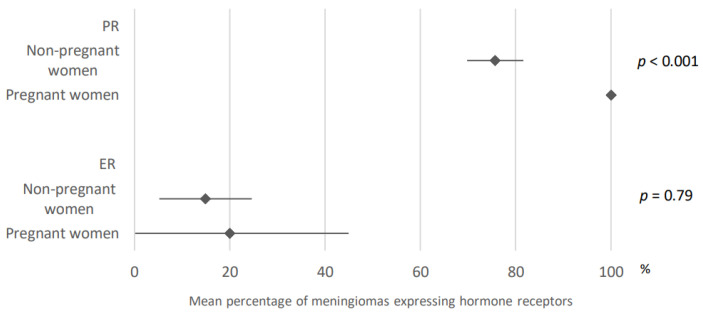
Comparison of hormone receptors expression within meningiomas developed in pregnant versus non-pregnant women. PR: progesterone receptors; ER: estrogen receptors. The results are expressed as a mean associated with their 95% confidence interval. Number of articles analyzed related to the topic: Progesterone receptors: Non-pregnant women: 27; Pregnant women: 5; Estrogen receptors: Non-pregnant women: 18; Pregnant women: 5; Number of cases analyzed expressed as total number of cases included (minimum number; maximum number): Progesterone receptors: Non-pregnant Women: 1940 (8; 396); Pregnant-Women: 27 (1; 17); Estrogen receptors: Non-pregnant Women: 1036 (15; 357); Pregnant-Women: 53 (1; 29).

**Figure 7 cancers-15-00980-f007:**
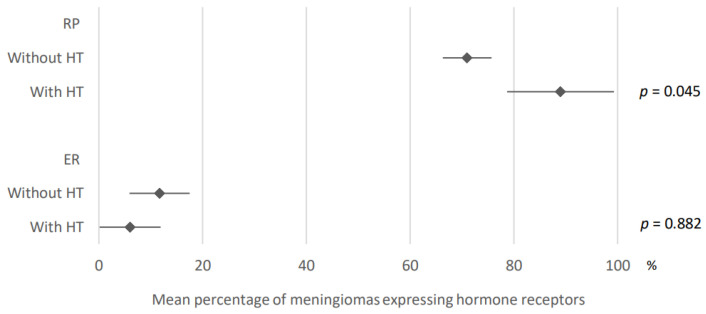
Comparison of hormone receptors expression in meningiomas developed under hormonal treatment versus without. PR: progesterone receptors; ER: estrogen receptors; HT: hormonal treatment. The results are expressed as a mean associated with their 95% confidence interval. Number of articles analyzed related to the topic: Progesterone receptors: without HT: 41; with HT: 3; Estrogen receptors: without HT: 24; with HT: 2. Number of included cases (total for all studies; minimum; maximum in each study): Progesterone receptors: without HT, 3838 (11; 588); with HT, 80 (24; 30); Estrogen receptors: without HT, 1472 (21; 161); with HT, 56 (26; 30).

**Figure 8 cancers-15-00980-f008:**
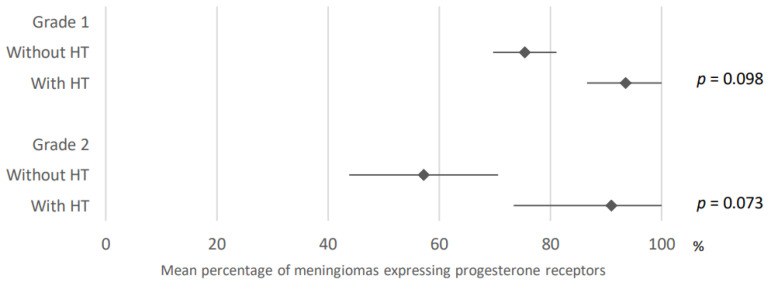
Comparison of progesterone receptors expression in meningiomas developed under and without hormonal treatment according to grade. HT: hormonal treatment. The results are expressed as a mean associated with their 95% confidence interval. Number of articles analyzed related to the topic: Grade 1: Without HT: 24; With HT: 2; Grade 2: Without HT: 15; With HT: 2; Number of cases analyzed expressed as total number of cases included (minimum number; maximum number): Grade 1: Without HT: 2500 (5; 533); With HT: 38 (11; 27); Grade 2: Without HT: 297 (2; 60); With HT: 6 (3).

**Figure 9 cancers-15-00980-f009:**
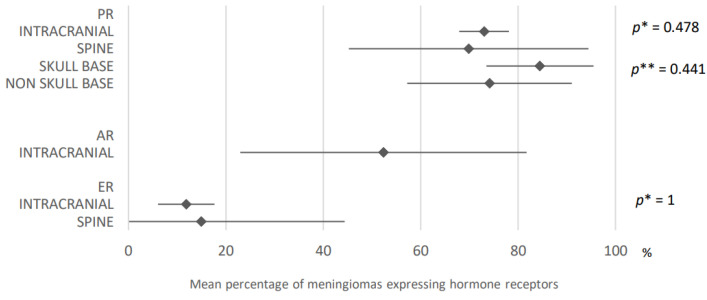
Comparison of hormone receptors expression in meningiomas according to their location. PR: progesterone receptors; AR: androgen receptors; ER: estrogen receptors. The results are expressed as a mean associated with their 95% confidence interval. *p**: intracranial versus spine; *p***: skull base versus non skull base. Number of articles analyzed related to the topic: Progesterone receptors: Intracranial: 36; Spine: 7; Skull base: 10; Non-skull base: 8; Androgen receptors: Intracranial: 5; Estrogen receptors: Intracranial: 24; Spine: 2; Number of included cases (total for all studies; minimum; maximum in each study): Progesterone receptors: Intracranial, 2685 (11; 588); Spine, 119 (1; 58); Skull base, 383 (4; 208); Non-skull base, 244 (7; 100); Androgen receptors: Intracranial, 246 (30; 96); Estrogen receptors: Intracranial, 1285 (11; 161); Spine, 34 (4; 30).

**Figure 10 cancers-15-00980-f010:**
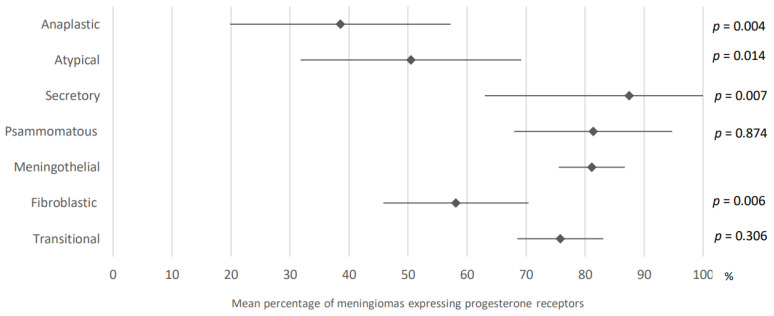
Comparison of progesterone receptors expression in meningiomas according to histological subtype. The results are expressed as a mean associated with their 95% confidence interval. *p*: meningothelial versus another histological subtype. Number of analyzed articles related to the topic: Anaplastic: 18; Atypical: 15; Secretory: 8; Psammomatous: 8; Meningothelial: 23 Fibroblastic: 20; Transitional: 22. Number of included cases (total for all studies; minimum; maximum in each study): Anaplastic, 57 (1; 7); Atypical, 50 (1; 10); Secretory, 69 (1; 31); Psammomatous, 81 (4; 28); Meningothelial: 935, (5; 376); Fibroblastic, 257 (1; 77); Transitional, 402 (3; 61).

**Figure 11 cancers-15-00980-f011:**
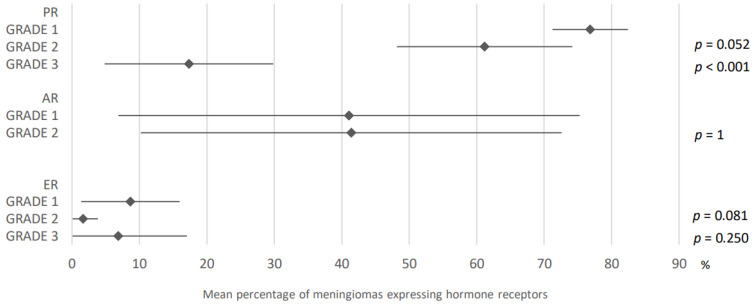
Comparison of hormone receptors expression in meningiomas according to grade. PR: progesterone receptors; AR: androgen receptors; ER: estrogen receptors. The results are expressed as a mean associated with their 95% confidence interval. *p*: grade 1 versus grade 2 or 3. Number of articles analyzed related to the topic: Progesterone receptors: Grade 1: 26; Grade 2: 17; Grade 3: 14; Androgen receptors: Grade 1: 4; Grade 2: 3; Estrogen receptors: Grade 1: 15; Grade 2: 10; Grade 3: 8. Number of included cases (total for all studies; minimum; maximum in each study): Progesterone receptors: Grade 1, 2538 (5; 533); Grade 2, 303 (3; 60); Grade 3, 164 (1; 87); Androgen receptors: Grade 1, 607 (27; 444); Grade 2, 53 (3; 39); Estrogen receptors: Grade 1, 1150 (11; 444); Grade 2, 154 (3; 60); Grade 3, 129 (1; 87).

**Figure 12 cancers-15-00980-f012:**
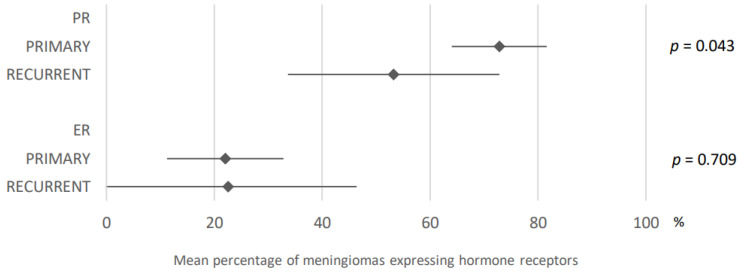
Comparison of hormone receptors expression in meningiomas according to their primary or recurrent nature. PR: progesterone receptors; ER: estrogen receptors. The results are expressed as a mean associated with their 95% confidence interval. Number of articles analyzed related to the topic: Progesterone receptors: Primary: 11; Recurrent: 9; Estrogen receptors: Primary: 8; Recurrent: 8. Number of included cases (total for all studies; minimum; maximum in each study): Progesterone receptors: Primary, 1839 (21; 588); Recurrent, 268 (1; 97); Estrogen receptors: Primary, 928 (21; 510); Recurrent, 153 (1; 67).

**Figure 13 cancers-15-00980-f013:**
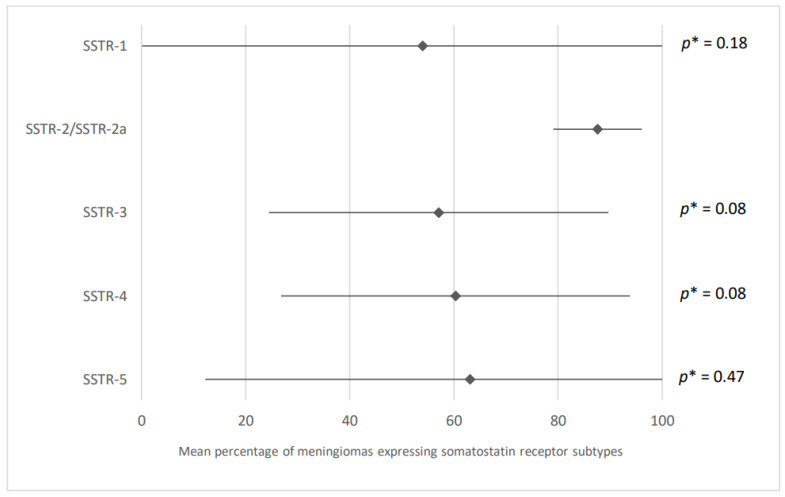
Expression of different somatostatin receptor subtypes in meningiomas. The results are expressed as a mean associated with their 95% confidence interval. SSTR: somatostatin receptor. *p**: subtype SSTR-2/SSTR-2a versus other subtypes. Number of articles analyzed related to the topic for each respective subtype: 3, 8, 3, 3, 3. Number of included cases (total for all studies; minimum; maximum in each study): SSTR-1/SSTR-3/SSTR-4/SSTR-5: 826 (40; 726); SSTR-2/SSTR-2a: 1122 (4; 726).

**Figure 14 cancers-15-00980-f014:**
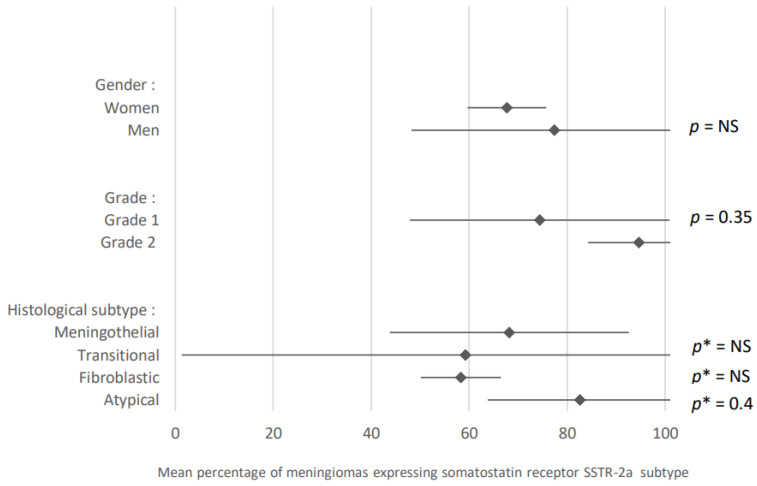
Comparison of somatostatin receptor SSTR-2a subtype expression in meningiomas according to different parameters. The results are expressed as a mean associated with their 95% confidence interval. *p**: meningothelial subtype versus other subtypes. NS: not significant. Number of articles analyzed related to the topic for each parameter: Gender: 2; Grade: 3; Histological subtype: 3. Number of included cases (total for all studies; minimum; maximum in each study): Gender: Women, 54 (22; 32)/Men, 21 (8; 13); Grade: Grade 1, 99 (13; 47)/Grade 2, 31 (1; 19) Histological subtype: Meningothelial, 47 (7; 29)/Transitional, 27 (1; 16) / Fibroblastic, 26 (2; 16); Atypical, 29 (3; 16).

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
