# Peer review of "Hormone Receptor Expression in Meningiomas: A Systematic Review"

_cancers, 2023, doi:10.3390/cancers15030980_

Round 1

Reviewer 1 Report

this is a very well prepared review article. The methodology of this article is noteworthy. The authors systematically presented the method and form of selection of the analyzed articles. The authors pointed out the exclusion of system errors in the analyzed articles. I have no comments on the methodological part of the article. The SUBJECT of the article is interesting. A particularly important message of the article is to draw attention to the therapeutic aspects and to verify the previous attempts at hormonal treatment. The authors selected only 3 types of receptors. It would be very interesting to supplement this review with results on somatostatin receptors.

Author Response

Reviewer 1

This is a very well-prepared review article. The methodology of this article is noteworthy. The authors systematically presented the method and form of selection of the analyzed articles. The authors pointed out the exclusion of system errors in the analyzed articles. I have no comments on the methodological part of the article. The SUBJECT of the article is interesting. A particularly important message of the article is to draw attention to the therapeutic aspects and to verify the previous attempts at hormonal treatment. The authors selected only 3 types of receptors. It would be very interesting to supplement this review with results on somatostatin receptors.

Thank very much for your comments. We interestingly supplemented our review with results on somatostatin receptors with the same method than the other receptors.

We added the following results, pages 12-14: “The SSTR-2 / SSTR-2a subtype is the most frequently expressed in meningiomas. On average, 87.6% of meningiomas expressed this latter (95% CI: 79.1; 96.1) versus 54% for the SSTR-1 subtype (95% CI: 0; 100), 57.1% for the SSTR-3 subtype (95% CI: 24.5; 89.7), 60.3% for the SSTR-4 subtype (95% CI: 26.8; 93.8) and 63.1% for the SSTR-5 subtype (95% CI: 12.2; 100). These results are presented in Figure 13. These results were obtained after analysis of 8 and 3 articles for the SSTR-2 / SSTR-2a subtype and for the others, respectively. Figure 14 summarizes the results of analyses comparing the expression of this SSTR-2/ SSTR-2a subtype in meningiomas according to different parameters. 67.7% of meningiomas developed in female patients expressed somatostatin receptor SSTR-2/ SSTR-2a subtype (95% CI: 59.7; 75.7) versus 77.4% of meningiomas in males (95% IC: 48.2; 100). These results were obtained after analysis of 2 articles. 74.4% of grade 1 meningiomas expressed somatostatin receptor SSTR-2/ SSTR-2a subtype compared to 94.7% of grade 2 (95% CI: 47.9-100 and 95% CI: 84.2-100). 3 articles dealt with this subject. Concerning the histological subtypes, 68.2% of meningothelial meningiomas expressed somatostatin receptor SSTR-2/ SSTR-2a subtype (95% CI: 43.8; 92.6) compared to 59.2% of transitional meningiomas (95% CI: 1.3; 100), 58.3% of fibroblastic meningiomas (95% CI: 50.1; 66.5) and 82.7% of atypical meningiomas (95% CI: 63.8; 100). These results were obtained after analysis of data from 3 articles.”

The Figures 13 and 14 were added accordingly (pages 13-14). Abstract (page 1; lines 18, 23, 28, 33-34), Introduction (page 3; lines 107-108), Materials and methods (page 3; lines 116-121) and PRISMA Flow Diagram (page 17; 62 references) were modified too. The data were shortly discussed (page 15; lines 528-531).

Reviewer 2 Report

The authors of this review reviewed relevant literature on the expression of progesterone, estrogen and androgen receptors in meningioma.  They looked at the expression of the receptors by age, sex, hormonal context, localization, histological subtypes, grade and recurrence.  

The data a interesting but several shortcomings are noted.  This is a retrospective study over 30 year period.  The methods used to investigate the different receptors has changed.  The antibodies have improved and the objective presence of receptors is not defined between studies.  All of these may make the results differ percentage wise.  Also, if immunohistological data is used for the presence of the receptors, antibodies become more important.  A negative ICC does not mean the receptor is not present necessarily.  

Some of the data is presented from as few as 2 studies.  That is a problem of bias.  The number of patients is each assessment is not presented from the data reviewed.  This may also bias the findings.  

The volume or site of the biopsy or tissue sample is not defined between studies.  This may also be important as the tissue is likely heterogenous as they are aware of.  

The data related to progesterone receptors is certainly the strongest for the 3 receptors.  The other two may also be important with a prospective study.  

Author Response

Reviewer 2

The authors of this review reviewed relevant literature on the expression of progesterone, estrogen and androgen receptors in meningioma. They looked at the expression of the receptors by age, sex, hormonal context, localization, histological subtypes, grade and recurrence. The data a interesting but several shortcomings are noted.

This is a retrospective study over 30-year period. The methods used to investigate the different receptors has changed. The antibodies have improved, and the objective presence of receptors is not defined between studies. All of these may make the results differ percentage wise. Also, if immunohistological data is used for the presence of the receptors, antibodies become more important.  A negative ICC does not mean the receptor is not present necessarily.

We agree with the reviewer. We had excluded for these reasons as well as to maintain homogeneity the oldest studies (before 1990), keeping only the studies evaluating the expression of receptors by immunohistochemistry. However, a negative immunostaining does not necessary mean that the receptor is not present. We have added these limitations in “Limitations of This Review” (page 14, lines 538-542).

Some of the data is presented from as few as 2 studies. That is a problem of bias. The number of patients is each assessment is not presented from the data reviewed. This may also bias the findings.

The number of patients in each assessment from the data reviewed was added throughout the manuscript, notably in each legend figures.

The volume or site of the biopsy or tissue sample is not defined between studies. This may also be important as the tissue is likely heterogenous as they are aware of.

The lack of data about the precise tumor localization and intra-tumoral heterogeneity with potential sampling bias have been added to the limitations in the discussion section, page 14, lines 536-538. Tissue sampling bias is on the other hand less important given that they are surgical specimens and not biopsies (page 3, line 131).

The data related to progesterone receptors is certainly the strongest for the 3 receptors. The other two may also be important with a prospective study.

Indeed, prospective studies are needed in order to confirm these results (added in discussion, page 14, line 542).

Round 2

Reviewer 2 Report

The authors have addressed my concerns regarding the studies and have adequately answered most of them in the discussion.  A new finding is the somatostatin receptor and this is adequately noted in figures and discussion.  No new concerns are noted.